# Confidence in a Vaccine against COVID-19 among Registered Nurses in Barcelona, Spain across Two Time Periods

**DOI:** 10.3390/vaccines10060873

**Published:** 2022-05-30

**Authors:** David Palma, Anna Hernández, Camila A. Picchio, Glòria Jodar, Paola Galbany-Estragués, Pere Simón, Montserrat Guillaumes, Elia Diez, Cristina Rius

**Affiliations:** 1Servei d’Epidemiologia, Agència de Salut Pública de Barcelona, 08023 Barcelona, Spain; ahernand@aspb.cat (A.H.); psimon@aspb.cat (P.S.); mguillau@aspb.cat (M.G.); ediez@aspb.cat (E.D.); crius@aspb.cat (C.R.); 2Consorcio de Investigació Biomèdica en Red en Epidemiología y Salud Pública (CIBERESP), 28029 Madrid, Spain; 3Department of International Health, Care and Public Health Research Institute—CAPHRI, Faculty of Health, Medicine and Life Sciences, Maastricht University, P.O. Box 616, 6200 MD Maastricht, The Netherlands; 4Barcelona Institute for Global Health (ISGlobal), Hospital Clínic, University of Barcelona, 08036 Barcelona, Spain; camila.picchio@isglobal.org; 5Col·legi Oficial d’Infermeres i Infermers de Barcelona, 08019 Barcelona, Spain; gjodar@coib.cat (G.J.); pgalbany@coib.cat (P.G.-E.); 6Department of Experimental and Health Sciences, Faculty of Health and Life Sciences, Universitat Pompeu Fabra (UPF), 08003 Barcelona, Spain; 7Institut de Recerca de l’Hospital de la Santa Creu i Sant Pau (IIB Sant Pau), 08041 Barcelona, Spain

**Keywords:** vaccine hesitancy, COVID-19, trust in vaccination, nurses, pandemic, epidemiology, safety concerns, beliefs in vaccination, vaccine recommendation

## Abstract

Objective: To report the vaccine hesitancy (VH) for a vaccine against COVID-19 in registered nurses in Barcelona, with measurements taken at two stages, prior to the vaccination campaign and once 75% vaccination coverage had been reached. Methods: A self-completed online survey was administered in December 2020 and again in July 2021 through the College of Nurses of Barcelona. It measured the prevalence of VH against a government-approved vaccine recommended by their employer, their intention to be vaccinated, perceptions of disease risk and vaccine protection, attitudes and beliefs to vaccination and social norm. Bivariate analysis according to VH and application time are presented. Results: 2430 valid responses were obtained in the first measurement and 2027 in the second. At both times, 86% were women and 69% worked mainly in the public sector. Prior to the vaccine availability, VH was 34.2%, decreasing to 17.9%. Risk perceptions were significantly lower in those with VH compared to non-VH, in all groups studied and at both times, while safety and efficacy perceptions increased in all groups, significantly less in VH. The greatest benefit of the COVID-19 vaccine is perceived by pharmaceutical companies. VH nurses perceived a more hesitant social environment. Conclusion: As the vaccination was rolled out, VH in nurses declined, with time improving the confidence in the safety and efficacy of the vaccines. Risk perceptions also decreased over time, except for the perception of severity in HCW where it increased. Trust in institutions impacts trust in vaccines.

## 1. Introduction

In 2019, it was predicted that vaccine hesitancy (VH) would be one of the 10 greatest threats to global health in the coming years [1]. Only a few months later, the apparition of the novel SARS-CoV-2 virus, which causes COVID-19 disease, fueled the debate on the importance and the need for a new vaccine, which was reliable, accessible and could be rapidly developed [2]. While the number of COVID-19 cases continues to rise, vaccine hesitancy has become a threat to effectively combatting this pandemic. The WHO defines VH as a continuum between doubt in acceptance and complete rejection, despite the availability of vaccination services [3,4]. It is a highly complex problem, dependent on the context, time, place, and vaccine in question [2,5].

Among the determinants in the decision of whether to be vaccinated, health professionals are recognized as constituting the most important influence [6,7,8], being the most trusted source of information among their patients. Nevertheless, some in the health community also present VH, which may impact on the uptake of vaccines by the general population [9]. Other factors that have contributed to VH in previous pandemics, epidemics and global outbreaks include demographic factors (ethnicity, age, sex, pregnancy, education, and employment); accessibility and cost; personal responsibility and risk perceptions; precautionary measures taken based on the decision to vaccinate; trust in health authorities and vaccines; the safety and efficacy of a new vaccine; and lack of information or vaccine misinformation [10].

Several studies have examined the COVID-19 vaccine VH in general population and healthcare workers (HCWs). A study of 13,426 people in 19 countries found a mean hesitancy of 28.5% [11], while a meta-analysis estimated a global COVID-19 vaccination willingness of 66.01% [95% CI: 60.76–70.89%] [12]. Age, gender, education, attitudes and perceptions about vaccines were the most frequently associated with vaccine acceptance or refusal in this meta-analysis. In HCWs, a systematic review reported extremely variable vaccine confidence rates according to territories [13]. One systematic review argued that, in addition to territories, ranging time points impacted considerably VH prevalence [14]. Another systematic review observed an increased willingness to vaccinate against COVID-19 in the US between 2020 and 2021 [15], arguing that timing of the survey should be an important measure when conducting such studies.

Maintaining confidence in vaccination is crucial to achieve an adequate vaccine coverage [16], so the challenges of this new vaccine are many. The development of different COVID-19 vaccines has advanced faster than any other vaccine before, raising questions about their efficacy and their ability to protect against COVID-19, as well as the safety of the development process [17]. In Barcelona, previous studies on general VH were performed in healthcare workers and demonstrated it be a growing issue of concern, which should be constantly evaluated [6]. In Spain, HCWs received the vaccination prior to the general population, so their VH could affect the recommendation to their patients. The aim of the study is to determine the level of VH in Barcelona nurses over the COVID-19 vaccines and to examine the factors associated with it, in two time points (prior to the approval of any vaccine, and after the local implementation).

## 2. Materials and Methods

### 2.1. Study Design and Participants

A descriptive cross-sectional study using a structured, online and self-administered survey was submitted in two stages. The first time point was at the beginning of the Spain’s national vaccination strategy, between 23 December 2020 and 25 February 2021. The second time point was between 15 June and 16 August 2021, when the COVID-19 vaccination campaign was already implemented, with 76.1% of Spain’s population and 81.8% of Catalonia’s population having received at least one dose by 25 August 2021 [18].

The questionnaire was applied to registered nurses in the province of Barcelona, in Catalonia, a region in the northeast of Spain. According to Catalonia’s Statistic Institute [19], as of 2020, there were 46,524 registered nurses, corresponding to 80.5% of the total number of registered nurses in Catalonia. Assuming an initial global prevalence of hesitancy of 28.5% [11], at least 925 participants were needed per recruitment, with a confidence of 95%, a precision of 3%, and an expected 15% loss.

### 2.2. Questionnaire and Data Collection

Researchers at the epidemiology department of Barcelona’s Public Health’s Agency developed the questionnaire using two previous instruments [6,11]. The survey was conducted in the QuestionPro platform and distributed with an invitation via email through the Barcelona’s College of Nurses’s database, to which the researchers did not have access. The link of the invitation could only be answered once, after agreeing with the informed consent, and only registered nurses from Barcelona received the email. Participants could stop answering the questionnaire at any time, so the dependent variable was asked in position 11 of 33 questions, excluding those who dropped out prior to that.

### 2.3. Variables

For the dependent variable, the question: “Would you accept a vaccine if it were recommended by your employer and was approved safe and effective by the government” [11] was categorized on a 5-point Likert scale with possible response options ranging from completely disagree to completely agree. Those who responded completely agree or some agreement were defined as not hesitant, and those who responded completely disagree, some disagreement, or neither agree nor disagree were defined as vaccine hesitant (VH). This information was crossed with an adaptation of a VH questionnaire on vaccines in the current vaccination schedule, carried out in the primary HCW of Barcelona [6]. The variables evaluated the risk perception associated with the disease; perception of vaccination; attitudes, beliefs, social norms; and sociodemographic variables.

The obtained variables were “Low risk perception” defined as impossible, unlikely or not probable or improbable, to the probability of contagion when being in contact with someone who has COVID-19, versus considering the contagion probable or very probable. The “Low severity perception” was defined as considering not serious, not very serious or moderately serious if infected, versus considering it serious or very serious. The “Low safety perception” was defined when selecting the vaccine as unsafe or dangerous versus considering it safe, very safe or totally safe. The “Low protection perception” was defined when considering the vaccine as moderately protective, not very protective or not at all protective, versus considering it protective or very protective. The “Low benefit perception” in determined groups of population was defined when selecting intermediate, little or “no benefit, versus considerably or a lot of benefit. Finally, when measuring attitudes, beliefs and social norm, the degree of agreement was dichotomized, based in an individual assessment, as hesitant or non-hesitant according to the meaning of every sentence.

Sociodemographic variables include year of birth, gender, type of practice (public, private or both), years of work and yearly family income. Due to the anonymous character of the questionnaire, personal data were not solicited at any moment. The questionnaire is available in Spanish and Catalan at confianzavacunaCOVID.questionpro.com, accessed on 20 April 2022.

### 2.4. Data Analysis

Univariate analyses are presented according to absolute and percentage frequency. For the quantitative variables, the mean and interquartile range (IQR) are presented. The bivariate analysis presents the dichotomized hesitancy according to the other variables, presenting measures of association in Chi 2 or student’s t, as appropriate. A *p*-value less than 0.01 was considered significant. Both the missing values and the Don’t Know/No answer (DK/NA) alternatives were analyzed separately but are presented together in Figure 3b for better understanding. All datasets were analyzed with STATA IC.16. The datasets are available in the repository of the Epidemiology Service of the ASPB, and the main findings are presented as Appendix A.

According to the Spanish Organic Law on Data Protection 3/2018 and the RGPD, informed consent was required for all participants prior to responding. Participation by those surveyed was voluntarily and could not be related to personal data.

## 3. Results

### 3.1. General Results

The first survey, administered at time point 1, obtained 2430 valid responses, while the second survey at point 2 obtained 2027 valid responses. Hesitancy in the first moment, measured according to presenting some disagreement, complete disagreement or neither agreement nor disagreement with the dependent variable, was 34.2%. At the second time point, vaccine hesitancy was 17.9%. Descriptive characteristics of the participant during both time points are presented in Table 1.

Figure 1 shows the differences between the agreement of a hypothetical vaccine, approved by the government and recommended by their employer [11], compared to the question on their actual uptake at the time of vaccinating [6]. It is observed that, at the first time point, some nurses who agreed to accept a recommended vaccine when asked in the correspondent item later in the survey respond to plan on delaying it or presents doubts or outright refusal to receive the vaccine. In the second time point, more nurses completely agreed on accepting the vaccine (from 895 to 1358 nurses), and of those, more had an actual intention or were already vaccinated (83.2% to 95.7%). Similarly, in the second time point, 52% (*n* = 92) of those who completely disagreed and 30.8% (*n* = 20) of those who slightly disagreed accepted the vaccine, without doubts or delay.

### 3.2. Risk Perception of the Disease

In the instance of coming in contact with a COVID-19 patient, the nurses considered both in the first and second time points that the highest risk of acquiring the disease occurs in patients or users in their daily practice (62.4% in the first sending and 53.3% in the second sending) and the lowest risk for those who live at home (52.2% and 45.8%, respectively) (Figure 2a).

There is a decrease in the perception of severity of an older adult and an adult with risk factors (81.8% and 85.8% in the first time point to a 54.3% and 71.1% in the second time point). There is an increase in the perception of severity of a healthcare professional contracting COVID-19 from the first to the second time point (41.2% to a 58.9%) (Figure 2b).

### 3.3. Perception of Vaccine Benefit

The perception of safety associated with the vaccine increases from 58.7% in the first time period to 84% in the second time period, significantly increasing both in the hesitant and non-hesitant groups (28.4% to 52.2% and 80.2% to 92%, respectively). The perception of protection increases from 37% to 62.3%, being significantly higher in the non-hesitant group in both time points (51.7% vs. 15.9% and 68.8% vs. 36.2%) (Figure 3a).

In the first moment, the highest perception of benefit was perceived in the pharmaceutical industry (67.1%) and the lowest perception of benefit was perceived in healthcare professionals (53.5%). In the second moment, the greatest benefit was perceived in older adults (74.2%) and the lowest benefit was perceived in the government (62.8%). In all groups and time points, the hesitant group perceived a significantly lower benefit than the hesitant group (Figure 3b).

### 3.4. Attitudes, Beliefs and Social Norm Related to Vaccination

There is a high agreement to rely more on vaccines which have been in use for longer periods of time, regardless of the degree of hesitancy or moment when the survey was administered. The reticent group presents misconceptions about vaccination that are persistent in both time points, doubling to little or no agreement when asked if vaccines improved every day thanks to research (6.4% for low or non-agreement in the first time point to 12.9% at the second moment), or that being vaccinated protects their nuclear group (13.8% to 26.5%) (Figure 4).

In relation to the social norm, there is a general perception increase in favor of vaccination in closest environments (68.8% to 83.8%) and a decrease in the perception of hesitancy in users of their daily practice (43.7% to 24.4%), decreasing both among those hesitant and non-hesitant. An increase in the perception of resources is also observed in the face of a situation of hesitancy (31.7% to 50.3%), being higher in the non-hesitant group (53% vs. 42.5%) (please refer to Appendix A for detailed tables).

## 4. Discussion

Our study shows that the prevalence of COVID-19 vaccine hesitancy among nurses in the province of Barcelona, Spain, decreased between December of 2020 to August 2021, decreasing the risk perceptions associated with the disease and increasing the perception of associated vaccine benefit. Prior to the implementation of the COVID-19 vaccination campaign, 34.2% of registered nurses in Barcelona had at least some doubts regarding the COVID-19 vaccine, decreasing to 17.9% as the vaccination campaign in Barcelona continued.

Hesitant nurses present lower risk perceptions associated with the disease than their peers who did not report hesitancy, and that perception becomes lower with the advancement of the vaccination strategy. This lower risk perception decreases significantly when asked about their patients and people who they co-habit with, potentially threatening the recommendation of COVID-19 vaccines. In our sample, nurses who perceived that their patients were less susceptible were less likely to recommend COVID-19 vaccines. Our results are in line with those of other studies which found vaccine acceptance to be higher among HCW who perceived themselves to be at greater risk or threat [20,21,22,23]. However, we found no significant differences in hesitancy according to contact or not with COVID-19 patients in their daily practice [24]. The perception of susceptibility could be modified as a result of highly contagious new strains, such as Omicron [25], although this variant had not been reported at the time of both submissions.

Hesitant nurses present a lower perception of severity than non-hesitant ones, which is twice as low for those who are an older adult and an adult with risk factors and decreases with the advance of the strategy. This finding is consistent with what has been reported in different countries [26], where hesitant HCWs have been described as having a consistently low severity perception, albeit being one higher than the general population [27]. Moreover, those who thought that a COVID-19 infection could be extremely severe in adults were 12.5 times more likely to have been vaccinated [28]. The expansion of vaccination and greater knowledge about patient management could have an impact on this perception globally, as French HCW have also been reported to exhibit a decrease in perception severity over time [29]. On the contrary, advancing the strategy, there was an observed increase in the perception of severity in HCW in both groups of nurses. Therefore, we can hypothesize that the progressive increase in professionals who have become ill could affect this perception, and at the same time contribute to the decrease in VH.

Regarding perceptions of the vaccine, safety and protection were seen to increase significantly in both groups with the time of submission, with a decrease in those who considered it dangerous, both perceptions being significantly lower in VH nurses. Safety and efficacy concerns have been described as the most common reasons for VH among general population and HCW within different territories. They have been reported as main concerns in studies in London [30], Oman [31], Hong Kong [32], Ethiopia [33], Istanbul [34], and Switzerland [35] among others. They have also been mentioned in different systematic reviews during the first year of pandemic, such as Al-Amer et al. [20], who targeted both as the main concern to address, especially in nurses, and Li et al. [22], who reported them as main barriers, together with distrust of the government. Safety concerns were the main reason for hesitancy in HCW from 37 countries [36], and nurses with confidence in safety reported an OR of 7.8 for vaccine uptake [37]. Only one study, performed in Italy [38], showed no association with safety concerns, but instead reported confidence in vaccine efficacy as the main predictor of vaccine uptake. At the same time, Xin et al. [39] reported that perceived vaccine efficacy mediated the effects of frequent social media exposure, increasing vaccine acceptance. It is possible that this efficacy perception diminishes with the need for new vaccines reinforcements to tackle new strains of the virus, for which constant vigilance and education about this perception are necessary.

Hesitant nurses perceived a significantly lower benefit from the vaccination in all the surveyed groups during both submissions. Nevertheless, while during the first submission, VH nurses perceived the lowest benefit in healthcare workers and the general community, the non-hesitant group perceived a lower benefit in the general community and the government. Although both agreed that the general community had been most affected by the pandemic, some hints of burn out and skepticism in the management of the epidemic by authorities could be represented prior to the vaccination strategy both in VH and non-VH nurses. Evidence of burn out in nurses was reported [40], yet we found no studies that correlated it with hesitancy, so it is something future interventions should take into account. When the vaccine strategy advanced, even when there is an increase in the overall perception of benefits, hesitant nurses continued to perceive the lowest benefit to be in the general community and healthcare workers, while the non-hesitant perceived the lowest benefit again to be in the government but also in the WHO. As a structural determinant of health [41], an institution’s response impacts the strengths of the healthcare system and, likewise, the perceptions of HCW. In that regard, one systematic review found low vaccine acceptance to be associated either with ineffective government efforts and initiatives [42] or with a lack of confidence in their management of the epidemic [22,33,35,43,44,45]. As well, trust in government is related to more confidence in vaccines [46,47].

The pharmaceutical industry, by contrast, was perceived by both groups, in both responses, to have the greater benefit. The only exception was the non-hesitance during the second time point, in which older adults were perceived as having benefited more. This change could be interpreted as a positive indicator of the vaccine strategy’s effectiveness. The lack of confidence in the pharmaceutical industry should be addressed because it could increase VH [44,45], especially in those who are hesitant, who completely reject the vaccine [48]. In our sample, the highest misconception in the hesitant group, during both times, was agreeing that vaccines are influenced by the illegitimate interest of the pharmaceutical industry. Although their contribution into the research and development of vaccines is undeniable, the prioritization of profit and ethical global distribution of vaccines is a challenge for pharmaceutical companies, governments, and international organizations [49].

Regarding attitudes and beliefs about vaccination, hesitant nurses had greater misconceptions than their non-hesitant peers, except when trusting in vaccines that had been used for a longer time of use, as opposed with the newest. In both times, non-hesitant nurses maintained a significantly lower or no agreement. This was a complex scenario with a very new vaccine developed in a relatively short period, so doubts over the time it had been in use are probably understandable. This concern was described in non-hesitant healthcare workers [48,50], but especially in those where lack of trust on science could be presented [33]. In our sample, hesitant nurses presented the lowest misconception during the first time point when referring to the fact that, thanks to research, vaccines are everyday better and safer, while during the second time, the lowest misconception was to trust vaccines with longer time of use.

Social norms play a critical role in shaping health-related behaviors and intentions [39]. During the first time point, almost twice as many of the hesitant nurses, compared with their non-hesitant peers, had a low or no agreement in believing that their close circle was in favor of being vaccinated, becoming almost three times higher during the second response. Previous studies conducted in Barcelona observed that hesitant HCWs also perceived people in their close environment to be significant among those less in favor of being vaccinated [6]. Given what has been discussed in terms of trust in institutions and political scenarios, it is important to combine the measures on individual perceptions with the community views on health systems so as to produce effective interventions [51].

Our study has certain limitations, such as the inability of determining causality due to the cross-sectional methodology. The voluntary online format may have led to a double selection bias of the more motivated nurses and those with better online skills. Yet, due to the size and characteristics of the sample, we believe it to be a representative sample of the study population, achieving some 4.4% of the total registered nurses of Barcelona in each time point. By receiving more than two times the expected responses in each submission, we consider our results to have a great power and to be highly generalizable for Barcelona’s reality. Moreover, the quantitative approach may not have captured other potential drivers of hesitancy. Some strengths of this study are its coordinated work with the College of Nurses, which has allowed us to expand our results to the region for future interventions. Similarly, the use of a dependent variable applied in many countries and populations has allowed much greater comparability. To consider the time as a variable is essential when monitoring this hesitancy, particularly with COVID-19 vaccines, whose rapid development time has marked a milestone in the history of vaccines. In this case, by administering the survey twice, we were able to identify characteristics during different stages of the pandemic, which should be useful in preparing for future needs. Other studies have evaluated variables, such as political identification or religiosity; however, the homogeneity of the responses in the first-time sample made it impossible to assess significant differences, so it was removed from the second instrument. Additionally, in the first questionnaire, one item made reference to the country where the vaccine originated, but in the second questionnaire, this was modified to the specific type of vaccine. We hope to present the rest of the data in a secondary analysis, focusing on the characteristics of delayers, doubters and complete refusals, and to design specific interventions for each group.

The implications of this study include the ability to better understand the hesitancy on previous regular vaccines, as well as to be prepared for future vaccines to come, especially those with similar characteristics, such as mRNA vaccines. The observed change in the hesitancy postulates that it is possible to decrease it with effective interventions. Together with this study, our team has been systematically reviewing the most effective interventions to address vaccine hesitancy in this specific population, to be published. With both studies finished, we will continue with an already obtained grant, to develop effective interventions to decrease hesitancy in healthcare workers in Barcelona.

## 5. Conclusions

In Barcelona, vaccine hesitancy surrounding a COVID-19 vaccine among nurses has decreased with time. VH depended on their perceptions of the benefits associated with the vaccine, as well as perceptions over risk related to the disease and attitudes and beliefs regarding general vaccination. The reasons for vaccine hesitancy were varied, with concerns related to the safety of the vaccine, and the follow-up time being the most commonly reported. The context impacted the trust in vaccination, and it was highly influenced by the government measures, so these should be taken into account in future studies.

The perceptions of risk between the first and second submissions have tended to converge. They decreased in the population at greatest risk but increased in healthcare workers. This may be something positive, speaking of the effectiveness of the management of strategies for COVID-19 vaccination, but at the same time, they expose the risk to which professionals are exposed. In relation to the perceptions of benefit, we consider the increase in the perceived benefit in older adults to be positive, but it is important to recognize the persistent and widespread distrust in institutions, such as the government, the pharmaceutical industry, and the World Health Organization. Maintaining trust in vaccines and the institutions involved in the entire development and administration process is essential to an effective global health response.

## Figures and Tables

**Figure 1 vaccines-10-00873-f001:**
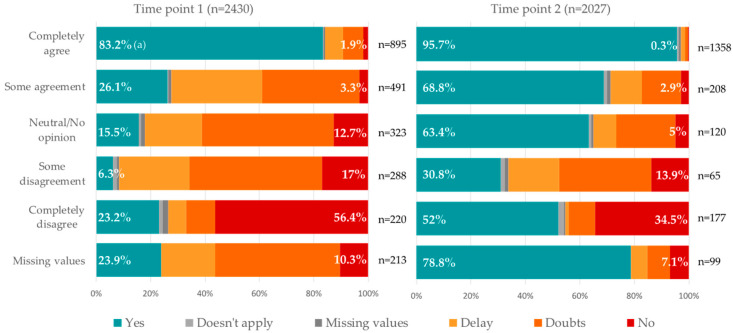
From intention to practice: Comparison between the agreement to accept an approved and recommended vaccine (Lazarus, ref. [11]) versus the actual intention to vaccinate (TP1) or already been vaccinated (TP2) (Picchio, ref. [6]); in Barcelona’s nurses, by submission time. COVID-19 Vaccine Hesitancy Study. Barcelona. 2020–2021. Percentage value at the left (green bar) belongs to “yes, they will get vaccinated/they have been vaccinated” answers, and values in the right (red bar) to “no, they won’t get vaccinated” answers. For values to “delay” and “doubts”, please refer to Appendix A.

**Figure 2 vaccines-10-00873-f002:**
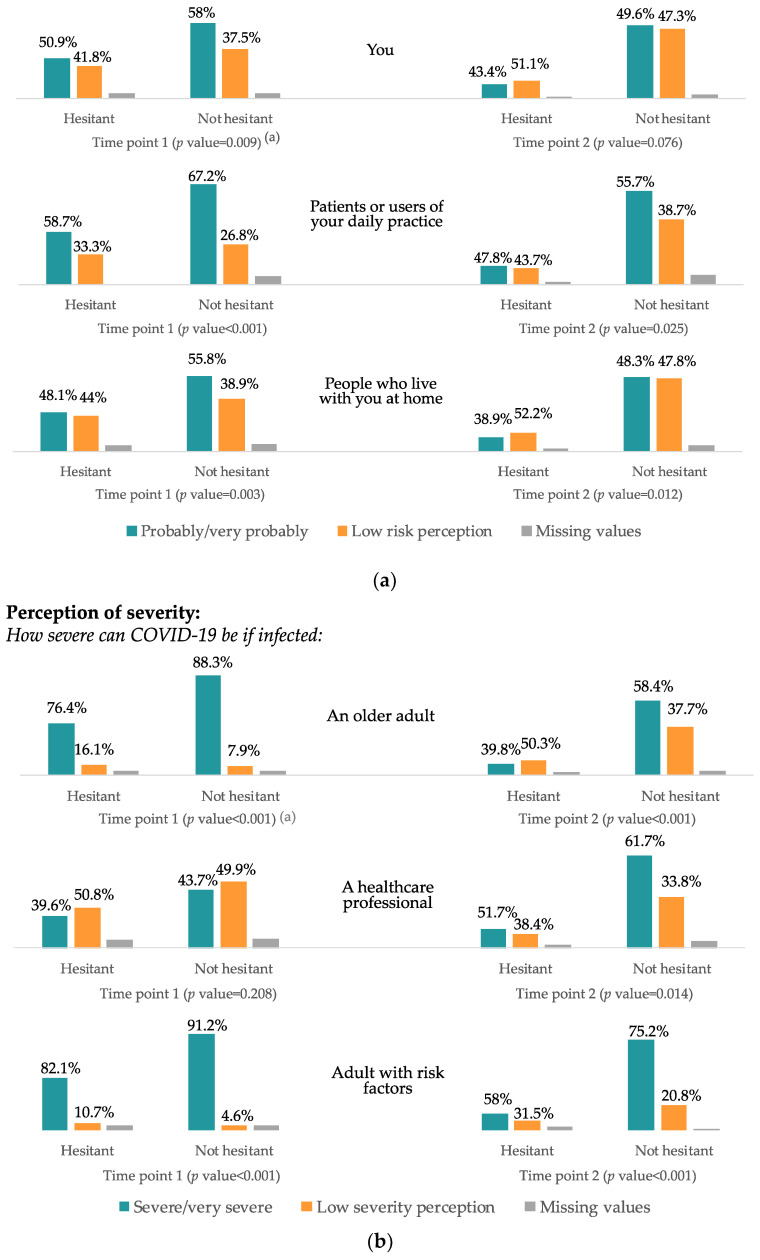
(**a**) Disease’s perception of risk: probability of infection, by hesitancy and time period. COVID-19 Vaccine Hesitancy Study. Barcelona. 2020–2021. (**a**) Chi square test between hesitant and non-hesitant nurses does not include missing values. Chi square test between time points were <0.001 for the three variables. Bar graphs present the absolute frequency between each group studied (you, patients or cohabitants) in both submissions, while the percentage frequency presents the risk perception separately in each group of nurses, according to VH or non-VH. (**b**) Disease’s perception of risk: perception of severity, by hesitancy and time point. COVID-19 Vaccine Hesitancy Study. Barcelona, 2020–2021. (^a^) Chi square test between hesitant and non-hesitant nurses does not include missing values. Chi square test between time points were <0.001 for the three variables. Bar graphs present the absolute frequency between each group studied (an older adult, a HCW or an adult with risk factors) in both submissions, while the percentage frequency presents the risk perception separately in each group of nurses, according to VH or non-VH.

**Figure 3 vaccines-10-00873-f003:**
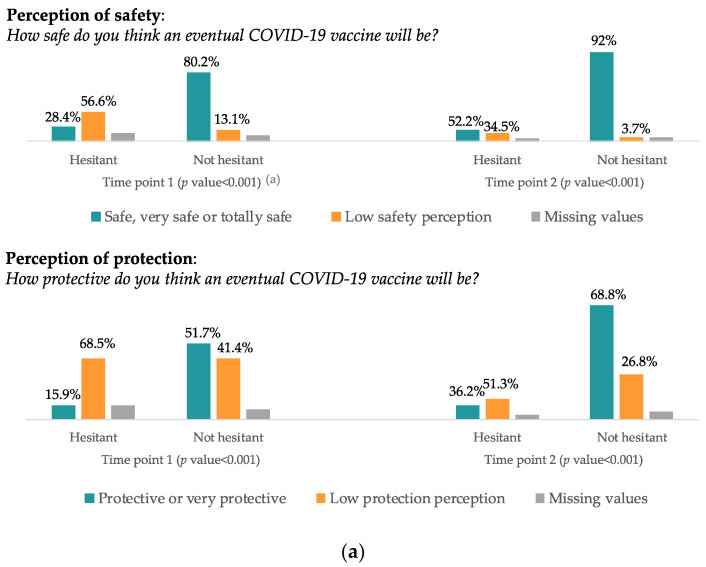
(**a**). Perception of vaccine benefit: safety and protection, by hesitancy and time point. COVID-19 Vaccine Hesitancy Study. Barcelona. 2020–2021^.(a)^ Chi square test between hesitant and non-hesitant nurses does not include missing values. Chi square test between submissions were <0.001 for both variables. Bar graphs present the absolute frequency between each group studied (safety or protection) in both submissions, while the percentage frequency presents the risk perception separately in each group of nurses, according to VH or non-VH. (**b**) Perception of vaccine benefit: benefit in specific populations, by hesitancy and time point. COVID-19 Vaccine Hesitancy Study. Barcelona. 2020–2021. (^a^) Chi square test between hesitant and non-hesitant groups was <0.001 in all comparisons and does not include missing values. Chi square test between submissions was <0.001 in all variables, except government (*p* = 0.009) and the pharmaceutical industry (*p* = 0.141). (^b^) H: Hesitant nurses, NH: Non-hesitant nurses. Missing values includes “Doesn’t know/No answer”.

**Figure 4 vaccines-10-00873-f004:**
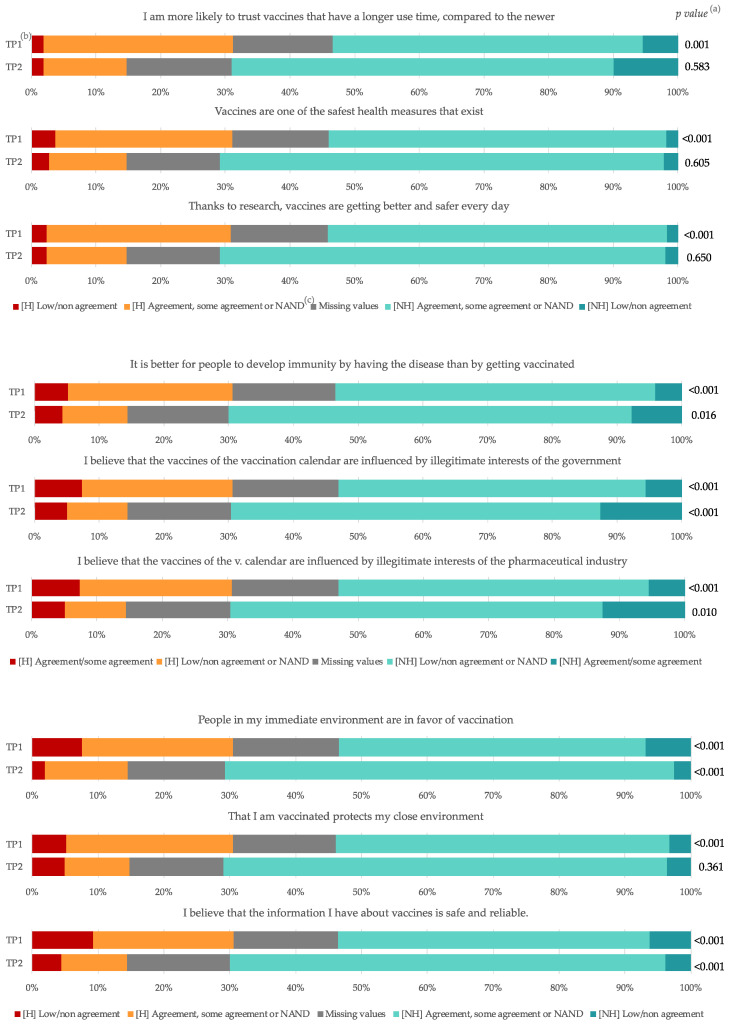
Attitudes, beliefs and social norm associated with general vaccination, by hesitancy and time point. COVID-19 Vaccine Hesitancy Study. Barcelona. 2020–2021. (^a^) Chi square test does not include missing values. (^b^) TP = Time points 1 and 2. (^c^) NAND: neither agreement nor disagreement. H: Hesitant nurses, NH: Non-hesitant nurses. Reticent options were placed in the extremes to better comparability between time points. They are defined according to the sense of the statement, so please check the legend below.

**Table 1 vaccines-10-00873-t001:** Descriptive analysis by submission time. COVID-19 Vaccine Hesitancy Study. Barcelona. 2020–2021.

	Time Point 1 XII 2020–II 2021 (*n* = 2430) ^a^		Time Point 2 VI–VIII 2021 (*n* = 2027) ^a^		*p* Value ^b^
Variable	*n*	%	*n*	%	
Age (Median + IQR)	43.4	42.8–44.0	45.5	44.8–46.1	0.0002
Years of work (Median + IQR)	18.2	17.7–18.7	20.2	20.2–21.4	<0.001
Gender					
Female	1596	86.37%	1381	86.37%	0.192
Male	242	13.32%	213	13.32%	
Other	1	0.05%	5	0.31%	
Type of practice					
Mainly public practice	1666	69.47%	887	68.71%	0.236
Mainly private practice	413	17.22%	248	19.21%	
Similar public and private practice	319	13.30%	156	12.08%	
Contact with COVID-19 patients					
Yes, COVID-19 patients as main task	763	31.45%	538	26.62%	0.002
Yes, but COVID-19(+) cases not as main task	437	18.01%	352	17.42%	
Yes, but COVID-19(+) cases are sporadic	769	31.70%	692	34.24%	
I have no contact with COVID-19(+) patients	426	17.52%	415	20.53%	
DN/NR	32	1.32%	24	1.19%	
Live with any dependents (minor under 14 years old, over 65 years old or sick people under care)					
Dependents	844	34.73%	739	36.46%	0.231
None of the above	1586	65.27%	1288	63.54%	
Has been infected with COVID-19					
Yes	508	21.18%	368	18.26%	0.015
No	1890	78.82%	1647	81.74%	
Severity if have been infected with COVID-19					
Mild	186	39.08%	120	33.52%	0.440
Moderate	260	54.62%	218	60.89%	
Severe	29	6.09%	19	5.31%	
Very severe	1	0.21%	1	0.28%	
Has been vaccinated against the flu this year					
Yes	1216	59.84%	835	62.45%	0.129
I haven’t/won’t be vaccinated	816	40.16%	502	37.55%	

^a^ Missing value are not presented. ^b^ Chi-square test between submissions does not include missing values.

## Data Availability

General data are provided in the Appendix A. For further information please write to David Palma (ext_dpalma@aspb.cat).

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
