# Peer review of "Confidence in a Vaccine against COVID-19 among Registered Nurses in Barcelona, Spain across Two Time Periods"

_vaccines, 2022, doi:10.3390/vaccines10060873_

Round 1
Reviewer 1 Report
The paper well photograph the vaccine hesitancy (VH) for an eventual vaccine against Covid-19 in registered nurses in Barcelona in December 2020 and in July 2021 to measure the prevalence of vaccine hesitancy (VH) against a government-approved vaccine recommended by their employer, their intention to be vaccinated, perceptions of disease risk and vaccine protection, attitudes and beliefs to vaccination and social norm. As the vaccination was rolled out, VH in nurses declined, with time improving the confidence in the safety and efficacy of the vaccines. The paper well describe the vaccine behavioral impact in the nurse category in Barcelona.
Author Response
Dear reviewer,
Thank you so much for your time and effort in this review. We hope to produce a useful manuscript, so we have taken in consideration your comments.
I'm attaching you our final manuscript with the comments of the three reviewers.
BW
David Palma and the research team.

Reviewer 2 Report
This is an interesting study on the significant vaccine-research topic.
Please consider the following revisions:
- It is unclear, why the Authors use the term "for an eventual vaccine against Covid-19" - In the reviewer's opinion, the word "eventual" can be removed. It will simplify the text and allow us to skip misunderstandings.
- Please provide more precise data on the questionnaire as well as the distribution of the questionnaire - line 86
- The figures are well-prepared and informative
- Line 249 - please avoid overwhelming sentences. "Our study shows that the prevalence of Covid-19 vaccine hesitancy among nurses
- decreased between December of 2020 to August 2021" - please specify that it refers to Barcelona city not the whole population of the Spanish nurses e.g. "vaccine hesitancy among nurses in Barcelona, Spain...."
- Please provide 2-3 sentences on the practical implications of this study
Author Response
Dear reviewer.
Thank you so much for your time and effort in this review. We hope to produce a useful manuscript, so we have taken in consideration your comments.
- It is correct that, at this moment (2022), the use of the word "eventual" creates some misunderstandings. When we first designed the project and submitted the first survey, there was still some doubts about which vaccine will be used in Europe, so "eventual" was used in the questionnaire. However, according to your suggestion it was removed in all the manuscript.
- According to your recommendations, we provided data on the questionnaire from lines 101 to 137.
- To provide more precise data on the questionnaire, we added in the line 139: "Sociodemographic variables include year of birth, gender, type of practice (public, private or both), years of work and yearly family income. Due to the anonymous character of the questionnaire, personal data wasn’t solicited at any moment. The questionnaire is available in Spanish and Catalan at: confianzavacunacovid.questionpro.com." As well, the authors are open to share the document as supplementary material if needed.
- To provide more precise data on the distribution, in the line 107 we have added: "The link of the invitation could only be answered once, after agreeing with the informed consent, and only registered nurses from Barcelona received the email"
- Thank you for your comments on the figures. We try to provide a great amount of information and at the same time being understandable. Tables with full datasets are provided as supplementary material.
- (and 5) According to your recommendations, it was added "nurses in the province of Barcelona, Spain,". We appreciate your recommendation to provide a better context in our results.
6. Practical implications were added in line 379: "The implications of this study include the ability to better understand the hesitancy on previous regular vaccines, as well as to be prepared for future vaccines to come, especially those with similar characteristics, like mRNA vaccines. The observed change in the hesitancy postulates that it is possible to decrease it with effective interventions. Together with this study, our team has been systematically reviewing the most effective interventions to address vaccine hesitancy in this specific population, to be published. With both studies finished, we will continue with an already obtained grant, to develop effective interventions to decrease hesitancy in health care workers in Barcelona"
We attached the final manuscript with comments of the three reviewers.
BW
David Palma and the research team

Reviewer 3 Report
Estimated Authors of the paper "Confidence in a vaccine against Covid-19 among registered nurses in Barcelona, Spain across two time periods",
I've read your paper with great interest. In this report, Palma et al. have performed a specific analysis about SARS-CoV-2 vaccine confidence on registered nurses from the area of Barcelona. Nurses were inquired in two specific time periods, i.e. december 2020 - february 2021; June-August 2021 (i.e. inception of the vaccination campaign among HCWs vs. late initial campaign, shortly before the introduction of booster doses for HCWs).
Nurses were recruited by means of the Barcelona's College of Nurses' database, allowing an extensive sharing of the questionnaire (as confirmed by the high number of collected questionnaire, that represent around 4-5% of potential recipients).
Eventually, Authors were able to collect the following information:
- vaccine hesitancy decreased from 34.2% (time point I) to 17.9% (time point II)
- Hesitant nurses presented a lower risk perceptions towards SARS-CoV-2 than their peers who do not report hesitancy, and that perception gets lower with the advancement of the vaccination strategy;
- Trust in vaccine safety increased in time point II vs. I
Collectively, such results may be particularly interesting for professionals involved in vaccination campaign (not only in HCWs), but some improvements are required to improve the readability of the paper:
1) Figure 1 should share the following information: "percentage of agreement between dependent variables", but it remains substantially unclear, at least from the point of view of the present reviewer. In Time Point 1, for example, what respondents were completely agreeing to? and so on. I would suggest Authors either to simplify the report or remove the fibure and summarize the share of individuals agreeing with vaccination in Table 1.
2) Figure 2a/b, 3a share information that are of substantial significance for the reader, but such information may be summarized in a table, that would be more simple to approach.
3) in the discussion, Authors should approach more extensively two substantial shortcomings of the paper:
a) the overall number of actual responders is relatively high, both in fact only 4-5% of potential participants were recruited. In other words, what about the potential generalizability of the results? Have you performed a preventive power analysis? In case, please include it. Otherwise, discuss the representitivity of the sample;
b) as the study was performed by means of a web based questionnaire, participants were potentially self-selected (i.e. oversampling of individuals that are more familiar with internet service, having a better attitude towards sharing personal information, etc) and again such issue should be discussed;
Eventually, please consider that ANNEX 1 was not available in my copy and therefore please provide the correspondent information in the revised version of this study.
Author Response
Dear reviewer.
Thank you so much for your time and effort in this review. We hope to produce a useful manuscript, so we have taken in consideration your comments.
- Indeed, after reviewing it we agree that is a bit difficult to understand. Sometimes the authors have read it so many times that misses some readability. Although this figure is different to the rest of the perception studied, provides a great information about the differences between the agreement to accept a vaccine versus the actual action to be vaccinated. However, the second one is not a dependent variable in this study, just creating confusion. The figure text was modified to: "From intention to practice: Comparison between the agreement to accept an approved and recommended vaccine (Lazarus, ref. 11) versus the actual intention to vaccinate (TP1) or already been vaccinated (TP2) (Picchio, ref. 6); in Barcelona’s nurses, by submission time. Covid-19 Vaccine Hesitancy Study. Barcelona. 2020-2021. Percentage value at the left (green bar) belongs to “yes, they will get vaccinated/they have been vaccinated” answers, and values in the right (red bar) to “no, they won’t get vaccinated” answers. For values to “delay” and “doubts”, please refer to supplementary data." Beside, in the results text (line 150) the paragraph was modified to: "Figure 1 shows the differences between the agreement of a hypothetical vaccine, approved by the government and recommended by their employer [11] compared to the question on their actual uptake at the time of vaccinating [6]. It is observed that, at the first time point, some nurses who agreed to accept a recommended vaccine when asked in the correspondent item, later in the survey respond to plan on delaying it or presents doubts or outright refusal to receive the vaccine. In the second time point, more nurses completely agree on accept the vaccine (from 895 to 1,358 nurses), and of those, more have an actual intention or have been already vaccinated (83.2% to 95.7%). Similarly, in the second time point, 52% (n=92) of those who completely disagreed and 30.8% (n=20) of those who slightly disagreed, accepted the vaccine, without doubts or delay."
- Thank you for your comments on the figures. We tried to provide a great amount of information and at the same time being understandable. Tables with full datasets are provided as supplementary material and I hope to get them better annexed now. As I hope you could see there, to present tables in two periods and between hesitant and non hesitants, plus the p values, created extremely big tables, that were decided to locate at the supplementary materials instead of in the manuscript.
- a. As you noted, the number of responses ir higher than expected. We performed an inverse calculation of power, obtaining over 100%, which is incorrect to add to the manuscript. Instead, we modified from line 359 as presented: "Yet, due to the size and characteristics of the sample, we believe it to be a representative sample of the study population, achieving some 4.4% of the total registered nurses of Barcelona in each time point. By receiving more than two times the expected responses in each submission, we consider our results to have a great power and to be highly generalizable for Barcelona’s reality."
- (3b). We initially discussed the selection bias just in the voluntary participation, but as you noted, it was a double selection bias. It was corrected from line 358: "The voluntary online format may have led to a double selection bias of the more motivated nurses and those with better online skills"
We attached the final manuscript with comments of the three reviewers.
BW
David Palma and the research team

Round 2
Reviewer 2 Report
The authors addressed all the comments.